# Ellagic Acid Affects Metabolic and Transcriptomic Profiles and Attenuates Features of Metabolic Syndrome in Adult Male Rats

**DOI:** 10.3390/nu13030804

**Published:** 2021-03-01

**Authors:** Adéla Kábelová, Hana Malínská, Irena Marková, Olena Oliyarnyk, Blanka Chylíková, Ondřej Šeda

**Affiliations:** 1Institute of Biology and Medical Genetics, The First Faculty of Medicine, Charles University and The General University Hospital, 121 08 Prague, Czech Republic; adela.kabelova@lf1.cuni.cz (A.K.); blanka.chylikova@lf1.cuni.cz (B.C.); 2Institute for Clinical and Experimental Medicine, 140 21 Prague, Czech Republic; haml@ikem.cz (H.M.); irma@ikem.cz (I.M.); ooliyarnyk@yahoo.com (O.O.)

**Keywords:** metabolic syndrome, ellagic acid, brown adipose tissue, insulin resistance, oxidative stress

## Abstract

Ellagic acid, a natural substance found in various fruits and nuts, was previously shown to exhibit beneficial effects towards metabolic syndrome. In this study, using a genetic rat model of metabolic syndrome, we aimed to further specify metabolic and transcriptomic responses to ellagic acid treatment. Adult male rats of the SHR-*Zbtb16^Lx^*^/k.o.^ strain were fed a high-fat diet accompanied by daily intragastric gavage of ellagic acid (50 mg/kg body weight; high-fat diet–ellagic acid (HFD-EA) rats) or vehicle only (high-fat diet–control (HFD-CTL) rats). Morphometric and metabolic parameters, along with transcriptomic profile of liver and brown and epididymal adipose tissues, were assessed. HFD-EA rats showed higher relative weight of brown adipose tissue (BAT) and decreased weight of epididymal adipose tissue, although no change in total body weight was observed. Glucose area under the curve, serum insulin, and cholesterol levels, as well as the level of oxidative stress, were significantly lower in HFD-EA rats. The most differentially expressed transcripts reflecting the shift induced by ellagic acid were detected in BAT, showing downregulation of BAT activation markers *Dio2* and *Nr4a1* and upregulation of insulin-sensitizing gene *Pla2g2a*. Ellagic acid may provide a useful nutritional supplement to ameliorate features of metabolic syndrome, possibly by suppressing oxidative stress and its effects on brown adipose tissue.

## 1. Introduction

Metabolic syndrome is defined as a simultaneous occurrence of several pathophysiological conditions including central obesity, dyslipidemia, hypertension, and impaired glucose tolerance that together greatly increase the risk for cardiovascular disease, type 2 diabetes, and other serious health conditions [1,2]. No single factor has yet been identified as an underlying cause of metabolic syndrome development, indicating that the pathogenetic mechanism is rather complex. Some evidence indicates that obesity, especially abdominal (visceral) obesity, plays a pivotal role in the process, being the most prevalent feature among patients with metabolic syndrome [3]. However, insulin resistance, chronic inflammation, and/or oxidative stress, which are typically accelerated in obese individuals, may also contribute to the progression of metabolic abnormalities related to the syndrome [4,5]. As the incidence of metabolic syndrome increases worldwide, it is important to establish simple and effective approaches to treat or possibly prevent this condition.

Ellagic acid is a naturally occurring polyphenolic compound found in numerous fruits, nuts, and seeds, such as pomegranates, grapes, raspberries, and walnuts, with antioxidative, antiobesity, hypolipidemic, and antidiabetic effects [6,7,8]. Either ellagic acid or urolithins, its metabolites, were repeatedly found to positively affect individual features of metabolic syndrome and its complications [6,9].

Concerning obesity, ellagic acid supplementation in rats fed a high-carbohydrate, high-fat diet attenuated body weight gain, lowered total abdominal fat deposition, and decreased abdominal circumference [9]. A suggested mechanism by which ellagic acid exerts its antiobesity effect includes suppression of energy intake and inhibition of the pancreatic lipase activity, which lead to reduced absorption of dietary lipids in the small intestine [10]. Other mechanisms, such as attenuation of fatty acid biosynthesis by inhibiting fatty acid synthase [11] or increasing fatty acid oxidation and consequently promoting their catabolism [12] upon ellagic acid treatment, were also described. 

In diabetic rats, ellagic acid decreased fasting blood glucose levels by increasing the size and number of pancreatic β-cells and stimulating insulin secretion [13]. In another study, the glucose tolerance was improved due to enhanced insulin signaling, which in turn led to the increased translocation of glucose transporter in the cell membrane and promotion of cellular glucose uptake [14]. Furthermore, ellagic acid was also able to attenuate diabetic retinopathy and nephropathy via inhibition of nonenzymatic glycation which is accelerated in hyperglycemic environments and contributes to the pathogenesis of long-term diabetic complications [15,16]. The increase in activity of antioxidative enzymes such as catalase and paraoxonase in nerve tissue of diabetic rats also indicates a neuroprotective effect of ellagic acid against oxidative damage [17]. 

Furthermore, supplementation of ellagic acid decreased plasma triacylglycerols and cholesterol levels in rats fed a high-fat fructose diet [14] and decreased LDL (low-density lipoprotein) and increased HDL (high-density lipoprotein) cholesterol levels in rats fed a high-fat diet [18]. The improvement in cholesterol metabolism may be partially due to the ability of ellagic acid to mediate endocytosis of LDL particles by upregulating LDL receptors content in liver tissue, and thus maintaining normocholesterolemia [19]. The development of atherosclerosis, endothelial dysfunction, and hypertension, risk factors for cardiovascular diseases, was also prevented by ellagic acid treatment [9,20,21].

In this study, we assessed the effect of acute oral administration of ellagic acid on metabolic parameters, oxidative stress, and transcriptomic profiles in an inbred rodent model carrying a variant of one of the metabolic syndrome-related genes, *Zbtb16* (zinc finger and BTB domain containing 16), known to have sensitizing properties to nutrigenetic and pharmacogenetic interactions and capable of also modulating some features of the metabolic syndrome [22,23].

## 2. Materials and Methods

### 2.1. Ethical Statement

All experiments were performed in agreement with the Animal Protection Law of the Czech Republic and were approved by the Ethics Committees of the First Faculty of Medicine of the Charles University and the Institute of Physiology, Czech Academy of Sciences, Prague (Permit Number: 66/2014).

### 2.2. Rat Strain

The SHR-*Zbtb16^Lx^*^/k.o.^ is a congenic rat strain that carries two distinct alleles of the *Zbtb16* gene on the spontaneously hypertensive rat (SHR/O1aIpcv, Rat Genome Database (RGD, ID: 631848) genomic background [24]. One of the two alleles of the *Zbtb16* gene was knocked out by TALEN (transcription activator-like effector nucleases) as described previously [25], while the other variant *Zbtb16* allele, together with seven other genes within the 788kb long segment of chromosome 8, is of polydactylous SHR-*Lx*.PD5 (SHR-*Lx*, RGD ID: 1641851) congenic rat strain origin [26]. Only the single Zbtb16 allele knockout model was used since the loss of function of both *Zbtb16* alleles is semi-lethal [25]. The derivation of the SHR-*Lx*.PD5 strain was described previously [27].

### 2.3. Experimental Protocol

Adult male rats of the SHR-*Zbtb16^Lx^*^/k.o.^ strain were held under temperature-(23 °C) and humidity-(55%) controlled conditions on a 12 h light/12 h dark cycle and fed a laboratory chow diet (STD, ssniff RZ, ssniff Spezialdiäten GmbH, Soest, Germany). At all times, the animals were given free access to food and water. At the age of 12 months, rats were randomly divided into two groups (*n* = 8/group), control group and experimental group, that were both fed a high-fat diet (HFD, ssniff EF R/M D12330 mod.—Surwit, ssniff Spezialdiäten GmbH, Soest, Germany) for three weeks. The experimental group was simultaneously supplemented with daily intragastric bolus gavage of 1.5 mL aqueous solution of ellagic acid (50 mg/kg body weight; high-fat diet–ellagic acid (HFD-EA) rats). The dose of ellagic acid used here was chosen to be similar to the doses used in previous studies [28]. The control group was supplemented with vehicle/solvent only (high-fat diet–control (HFD-CTL) rats). Body weight and food intake were measured twice a week for both groups.

Blood samples for metabolic and glycemic assessments were drawn after overnight fasting (12 h) from the tail vein. For the oral glucose tolerance test, blood samples were obtained prior to and at intervals of 30, 60, 90, 120, and 180 min after intragastric glucose administration to conscious rats (3 g/kg body weight, 30% aqueous solution). Blood glucose concentrations over the period of 180 min were used to calculate the area under the curve. Subsequently, all rats were sacrificed and the weights of heart, liver, kidneys, adrenals, muscle, and brown, epididymal, and retroperitoneal adipose tissue were determined. All organs were snap-frozen in liquid nitrogen and stored at −80 °C for further analysis. Serum insulin concentrations determined by using a rat insulin enzyme-linked immunosorbent assay kit (Mercodia, Uppsala, Sweden) and alanine aminotransferase (ALT) and aspartate aminotransferase (AST) enzyme activities determined spectrophotometrically by routine clinical biochemistry methods using a kit from Roche Diagnostic (Germany) were assessed at the Institute for Clinical and Experimental Medicine, Prague, Czech Republic.

### 2.4. Lipid Profile Assessment

The serum lipid profile was assessed using gel-permeation high-performance liquid chromatography at Skylight Biotech Inc. (Akita, Japan) allowing simultaneous quantification of cholesterol and triacylglycerols in 20 lipoprotein subfractions as described previously [29,30]. In short, a three-step method was utilized, including the loading of sample into a gel permeation column followed by elution of lipoproteins in decreasing order of size, feeding them into reaction coils; the products were registered by detectors and the levels of triacylglycerols and cholesterol in the individual lipoprotein fractions were obtained in the form of composite chromatogram. The level of nonesterified fatty acids was determined by using a kit from Roche Diagnostic (Germany).

### 2.5. Oxidative Stress Markers Assessment

Liver, kidney, and heart tissue were quickly homogenized with Potter–Elvejhem homogenizer at 0–4 °C to proceed with analysis of oxidative stress markers. The levels of reduced and oxidized forms of glutathione were determined by using a high-performance liquid chromatography method with fluorescent detection according to the HPLC diagnostic kit (Chromsystems, Munich, Germany). The activities of antioxidant enzymes superoxide dismutase, glutathione peroxidase, glutathione reductase, and glutathione S-transferase were determined by using commercially available assay kits (Sigma-Aldrich and Cayman Chemicals; Ann Arbor, MI, USA). Catalase activity measurement was based on the ability of H_2_O_2_ to produce a color complex with ammonium molybdate detected spectrophotometrically. Lipoperoxidation products were assessed according to levels of thiobarbituric acid reactive substances (TBARS) determined by assaying the reaction with thiobarbituric acid [31,32].

### 2.6. Transcriptomic Analysis

Total RNA was isolated from liver and brown and epididymal adipose tissue (RNeasy Mini Kit, Qiagen, Hilden, Germany). The quality and integrity of the total RNA were evaluated on an Agilent 2100 Bioanalyzer system (Agilent, Palo Alto, CA, USA) and only samples with RNA Integrity Number (RIN) >8.0 were utilized in further steps of the protocol. Microarray experiments were performed using the Rat Gene 2.1 ST Array Strip in quadruplicate for each group/tissue combination (i.e., a total of 24 arrays were processed). The hybridization procedure was performed using the Affymetrix GeneAtlas^®^ system according to manufacturer’s instructions. The quality control of the chips was performed using Affymetrix Expression Console software (Affymetrix, Santa Clara, CA, USA). Partek Genomics Suite (Partek, St. Louis, MO, USA) was used for subsequent data analysis. After applying quality filters and data normalization by the Robust Multichip Average (RMA) algorithm, the set of obtained differentially expressed probe sets was filtered by the false discovery rate (FDR < 0.05) method that is implemented in Partek Genomics Suite 7 (Partek, St. Louis, MO, USA).

Transcriptomic data were then processed by standardized sequence of analyses (hierarchical clustering and principal component analysis, gene ontology, gene set enrichment, ‘Upstream Regulator Analysis’, ‘Mechanistic Networks’, ‘Causal Network Analysis’ and ‘Downstream Effects Analysis’) using Ingenuity Pathway Analysis (Qiagen). The microarray data generated and analyzed during the current study are available in the ArrayExpress repository (https://www.ebi.ac.uk/arrayexpress/) under accession number E-MTAB-8928.

### 2.7. RT-qPCR

To validate the gene expression data obtained by microarray, quantitative real-time PCR (RT-qPCR) was performed. The amount of 1 µg of total RNA was used to synthesize cDNA using oligo-dT primers and the SuperScript III reverse transcriptase (Invitrogen, Carlsbad, CA, USA). For validation, the following sets of TaqMan^®^ probes (Thermofisher; Waltham, MA, USA) were used: deiodinase, iodothyronine, type II (*Dio2*): Rn00581867_m1, glucokinase (*Gck*): Rn00561265_m1, phospholipase A2, group IIA (platelets, synovial fluid) (*Pla2g2a*): Rn00580999_m1, tenascin XB (*Tnxb*): Rn01526063_m1, nuclear receptor subfamily 4, group A, member 1 (*Nr4a1*): Rn01533237_m1. RT-qPCR reaction was performed in triplicate with TaqMan Gene Expression Master Mix (Applied Biosystems) according to the manufacturer’s protocol (Invitrogen, Carlsbad, CA, USA) using Applied Biosystems 7900HT Real-Time PCR System. Cycle threshold (Ct) values were normalized by using glyceraldehyde-3-phosphate dehydrogenase (Gapdh) (TaqMan^®^ chemistry, Applied Biosystems) as standard. Relative quantification was performed using the Livak method [33].

### 2.8. Statistical Analysis

All statistical analyses were performed in Statistica (data analysis software system), version 13.5 (TIBCO Software Inc., Palo Alto, CA, USA). Morphometric and metabolic variables of the two groups were compared by unpaired Student t-test where *p*-value < 0.05 was considered significant.

## 3. Results

### 3.1. Morphometric and Metabolic Parameters

The effect of ellagic acid on morphometric and metabolic parameters in HFD-EA and HFD-CTL rats is summarized in Table 1. No differences in the food intake and total body weight of rats during the experimental period were observed between the two groups. The relative weight of brown fat was increased while that of epididymal fat was decreased in HFD-EA rats compared to HFD-CTL rats. No significant differences in retroperitoneal fat mass and weights of other organs were detected between the two groups (Table 1).

Fasting blood glucose concentrations were significantly lower in rats supplemented with ellagic acid. During the oral glucose tolerance test, lower blood glucose concentrations at 90th and 120th minutes, along with smaller area under the curve, were detected in HFD-EA rats compared to HFD-CTL rats (Figure 1a and Figure 2b). Furthermore, ellagic acid supplementation decreased levels of fasting insulin (Figure 1c).

Serum free glycerol and total cholesterol concentrations were lower in rats supplemented with ellagic acid (Table 1). The decrease in cholesterol content of lipoprotein fractions was predominantly observed within the small and very small LDL particles (Figure 2a). No significant differences were detected in nonesterified fatty acids, total triacylglycerols concentrations (Table 1), and the triacylglycerol content of the individual lipoprotein fractions (Figure 2b) between the HFD-CTL and HFD-EA groups (Table 1). The activity of alanine aminotransferase and aspartate aminotransferase did not differ between the two groups (Table 1).

### 3.2. Oxidative Stress Markers

As shown in Figure 3, ellagic acid treatment markedly reduced oxidative stress in the liver tissue. The ratio of reduced/oxidized glutathione was significantly higher in the liver in HFD-EA rats in comparison to HFD-CTL rats (Figure 3a). In addition, ellagic acid supplementation increased superoxide dismutase, catalase, and glutathione reductase activities (Figure 3b–d) and decreased the concentration of TBARS in the liver tissue of HFD-EA rats (Figure 3e). Only minor changes in oxidative stress markers were observed in the kidney and heart tissue between HFD-EA and HFD-CTL rats (Appendix A).

### 3.3. Transcriptomic Analysis

When comparing HFD-CTL and HFD-EA rats, we detected 121 differentially expressed transcripts in brown adipose tissue. In contrast, no differentially expressed transcripts in liver and epididymal fat passed the significance threshold when corrected for multiple comparisons (false discovery rate (FDR) < 0.01). The summary of all significantly differentially expressed transcripts in brown adipose tissue is shown in Appendix A. The genes showing the most substantial decrease in expression by ellagic acid administration included iodothyronine deiodinase 2 (*Dio2*), acyl-CoA thioesterase 11 (*Acot11*), or PPARG coactivator 1 alpha (*Ppargc1a*). At the same time, angiopoietin-like 1 (*Angptl1*), stathmin 2 (*Stmn2*), or 5-hydroxytryptamine receptor 2A (*Htr2a*) belonged to the most upregulated ones. The analysis of potential upstream regulators based on the complete dataset of differentially expressed transcripts revealed predicted inhibition of LDL and fatty acid synthesis along with predicted activation of lysine demethylase 5A (*Kdm5a*) and tumor protein P53 (*Tp53*) (Appendix A). Figure 4 summarizes the results of network analysis, suggesting *Ppargc1a* together with nuclear receptor subfamily 4 group A members 1, 2, and 3 (*Nr4a1, Nr4a2, Nr4a3*) as major nodes forming the central module underlying the main EA effect on brown adipose tissue. The selected transcripts validated by qPCR are summarized in Appendix A.

## 4. Discussion

Metabolic syndrome is a cluster of risk factors, such as obesity, impaired glucose tolerance, hypertension, and dyslipidemia, that greatly increase the chance of developing cardiovascular disease, type 2 diabetes, or cancer. Since the prevalence of metabolic syndrome is rising, it is important to establish effective strategies to treat or even prevent this serious health condition. Ellagic acid, a phytochemical found in various fruits and nuts, has been previously shown to improve obesity and glucose tolerance, in vitro and in vivo, showing promising results against metabolic syndrome [9,28,34]. In this study we investigated the effect of ellagic acid on metabolic parameters, including oxidative stress markers as well as transcriptomic profile in a genetic rat model of metabolic syndrome.

Although the exact pathogenetic mechanism of metabolic syndrome development remains to be elucidated, it is believed by some that abdominal (visceral) obesity plays a major role in the process, being the most prevalent feature in patients with this condition [35]. Ellagic acid has been previously shown to counteract obesity via inhibiting adipogenesis, de novo lipogenesis, and pancreatic lipase activity, and/or enhancing catabolism of fatty acids, thus decreasing abdominal fat deposits and abdominal circumference [9,10,12]. Likewise, in our study, a lower relative weight of epididymal fat (a visceral adipose tissue) in rats supplemented with ellagic acid (HFD-EA rats) was observed, interestingly, without any changes in food intake or total body weight between the two tested groups. This observation, however, seems to be in agreement with some previous studies indicating that the effect of ellagic acid on body weight gain and adiposity may differ and depend not only on the source and content of ellagic acid, but also on the supplementation duration [6,28,36].

Increased intra-abdominal fat deposition is also typically associated with insulin resistance, dyslipidemia, and oxidative stress, which further promote metabolic disturbances [37,38,39]. As expected, we detected significantly lower levels of fasting insulin and glucose along with the area under the curve following the oral glucose tolerance test in HFD-EA rats compared to HFD-CTL rats. These results corroborate other findings demonstrating strong hypoglycemic properties of ellagic acid in diabetic animals [9,28,40]. Furthermore, some evidence also suggests a beneficial effect of ellagic acid in preventing the development of long-term diabetic complications, such as retinopathy, nephropathy, and neuropathy [15,16,17]. However, the exact mechanism by which ellagic acid exerts its antidiabetic effect remains unknown. A possible mechanism could be explained by decreasing serum resistin level, which was observed in some previous studies after ellagic acid treatment [41,42]. Resistin is a hormone secreted by adipocytes whose level positively correlates with insulin resistance and is thus thought to be the link between obesity and type 2 diabetes [41,42]. However, as we did not measure the resistin concentration directly and its expression in the adipose tissue was not different between the control and experimental group, its possible involvement remains to be confirmed. Other mechanisms, such as stimulating insulin secretion by improving the function of pancreatic β-cells [13] or enhancing insulin signaling that in turn results in increased cellular glucose uptake, were also proposed [14].

Ellagic acid also possesses the ability to normalize blood lipid levels, specifically, a decrease in levels of triacylglycerols, total cholesterol, and LDL cholesterol, and an increase in levels of HDL cholesterol [14,18]. The mechanisms involved in cholesterol normalization after ellagic acid administration include upregulation of LDL receptors in liver tissue [19] and/or promotion of cholesterol removal through increasing bile acid excretion [43]. Consistent with these findings, we detected a decreased total cholesterol level, predominantly in the LDL subfraction. However, the differences in levels of TG between HFD-EA and HFD-CTL rats did not reach statistical significance. Other studies also failed to confirm the TG-lowering properties of ellagic acid, with some even reporting increased serum TG level upon ellagic acid administration [8,28]. A possible explanation for these diverse results may be due to longer supplementation duration and/or different source of ellagic acid that was used in some previous studies [6,44]. Further studies are needed to fully elucidate the effect of ellagic acid on blood lipids levels.

Metabolic syndrome is also typically associated with permanently increased oxidative stress that is caused by an increase in concentration of reactive oxygen species (ROS). It has been shown that the susceptibility to oxidative damage is greater in obese subjects due to depletion of antioxidant enzymes, such as superoxide dismutase, glutathione peroxidase, or catalase, that neutralize ROS [45]. Among the three tested tissues, we observed the greatest reduction of oxidative stress markers in the liver, compared to kidneys and heart tissue. The activity of antioxidant enzymes glutathione reductase, superoxide dismutase, and catalase was increased in the liver of HFD-EA rats, together with a decrease in the level of TBARS, a marker of lipid peroxidation caused by ROS. Moreover, the ratio of reduced and oxidized glutathione (GSH:GSSG) that inversely correlates with the level of oxidative stress and hepatic lipid accumulation was increased in the liver of HFD-EA rats. As a result of its antioxidant effects, ellagic acid can thus possibly improve hepatic functions and prevent the development of nonalcoholic fatty liver disease (NAFLD), a hepatic disorder often accompanying metabolic syndrome [46,47]. The beneficial effect of ellagic acid against increased oxidative stress, particularly in the liver, was further confirmed in other studies with diabetic, hypertensive, or hyperlipidemic animal models [28,46,47]. Surprisingly, the level of TBARS was decreased in heart tissue of HFD-EA rats, although no changes in antioxidant enzyme activity and GSH/GSSG ratio were detected. It has been previously reported that besides increasing the activity of antioxidant enzymes, ellagic acid also exhibits antioxidant activity on its own as the molecule of ellagic acid contains two lactone groups that can act as donor and acceptor of hydrogen bonds and thus be involved in scavenging ROS and decreasing lipid peroxidation [48,49].

While abdominal fat (which is considered a white adipose tissue) affects metabolic parameters rather unfavorably, brown fat seems to have a protective effect in this regard. Brown fat is specialized to maintain body temperature through nonshivering thermogenesis and thus protect mammals from hypothermia [50]. Only recently, an ability of ellagic acid to promote browning of white adipose tissue was established [51,52]. This alteration of white fat leading to an increase in energy expenditure upon metabolic activation of brown fat is making ellagic acid a new potential target for treating obesity and obesity-related diseases, such as metabolic syndrome. Several genes specifically expressed in white fat during the browning process induced by ellagic acid have been detected. These include: uncoupling protein 1 (*UCP1*), PR domain zinc finger protein (*PRDM16*), peroxisome proliferator-activated receptor gamma coactivator 1-alpha (*PGC1α*), t-box protein 1 (*TBX1*), and nuclear respiratory factor 1 (*NRF1*); these are primarily involved in the production of fatty acid serving as a fuel for heat generation in mitochondria [51,52]. To our knowledge, there is so far no information available on ellagic acid effects on brown adipose tissue transcriptome. Although the results of our study did not confirm the ability of ellagic acid to induce browning of white fat, a higher relative weight of brown fat was observed in EA-treated animals compared to controls. The most ellagic acid-upregulated gene in our model, the phospholipase A2 group IIA (*Pla2g2a*), was previously implicated in metabolic regulation [53]. Humanized C57BL/6 mice expressing the human *PLA2G2A* gene were more insulin sensitive and glucose tolerant compared to C57BL/6 mice, both on standard and high-fat diet [54] and showed lower HDL and higher LDL cholesterol levels [53]. The seemingly counterintuitive results of gene expression comparison showed a concurrent, substantial decrease in brown adipose tissue-enriched gene *Ppargc1a* as well as of brown adipose tissue activation markers *Dio2* and *Nr4a1*. Another substantially downregulated gene in ellagic acid-treated rats was the acyl-CoA thioesterase 11 (*Acot11*). *Acot11*-deficient mice show resistance to diet-induced obesity and diabetes [55], however, contrasting, strain-specific effects of this brown adipose tissue-specific enzyme were observed in response to high-fat diet feeding [56]. Therefore, the brown adipose tissue does not seem to be metabolically activated by ellagic acid in the SHR-*Zbtb16^Lx^*^/k.o.^ strain but rather used as a preferential excess fat depot. This may result from a strain-specific reaction, perhaps connected to the semi-deficient variant *Zbtb16* allele. Overexpression of *Zbtb16* in brown adipocytes was shown to enhance the expression of several genes within the thermogenic program and higher levels of *Zbtb16* expression correlated with decrease in body fat storage [22,57].

The limitations of the current study include the use of only adult male rats of a single inbred strain as sex-specific genetic architecture of metabolic syndrome [58]. Also, the design of our experiment did not include the group of rats fed a standard chow diet and therefore does not show the extent of negative effects of the high-fat diet or the benefits provided by the ellagic acid in this perspective. Single dose and regimen of ellagic acid administration in an inbred strain that was used in the study does not allow an assessment of the dosage-dependent effects and potential nutrigenetic and pharmacogenetic aspects of ellagic acid effects. The samples for oxidative stress assessment were processed on ice without glutathione oxidation protectors, which may have affected the GSH:GSSG ratio as an indicator of oxidative stress. As we specifically selected an inbred model carrying alleles repeatedly shown to greatly sensitize their carrier to dietary and pharmacologic interventions [27], only a three-week exposure to HFD was selected.

In conclusion, we show that an EA-fed inbred model carrying a variant of *Zbtb16* gene shows better glucose tolerance, lower amount of visceral adipose tissue, lower cholesterol and insulin, and reduced measures of oxidative stress compared to its genetically identical control. The higher relative amount of brown adipose tissue in EA-treated rats was accompanied by a shift in its transcriptomic profile showing downregulation of BAT activation markers such as *Dio2* and *Nr4a1* and upregulation of insulin-sensitizing gene *Pla2g2a*. As the prevalence of metabolic syndrome is still increasing, the dietary supplementation with either purified ellagic acid or food high in ellagic acid could potentially be an effective interventional strategy.

## Figures and Tables

**Figure 1 nutrients-13-00804-f001:**
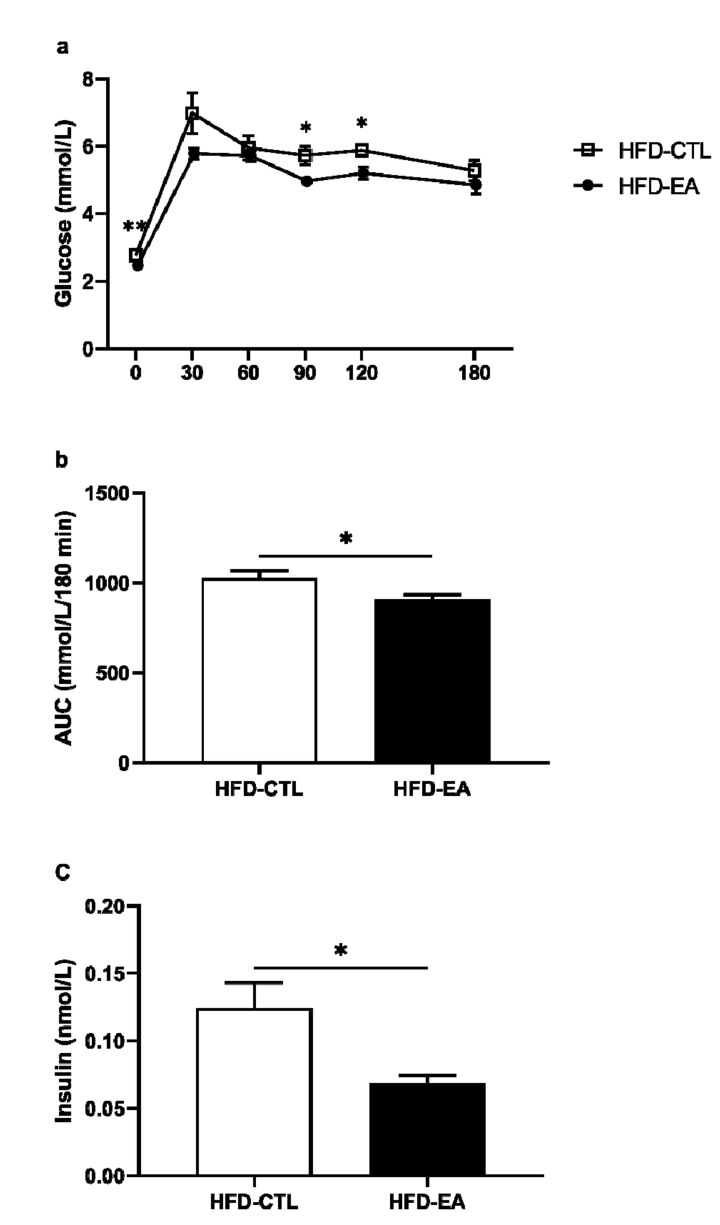
Effect of ellagic acid supplementation on blood glucose levels during (**a**) oral glucose tolerance test, (**b**) area under the curve (AUC), and (**c**) insulin levels. Values are mean ± SEM, n = 8/group. The significance levels for comparison between control (HFD-CTL; high-fat diet-fed rats) and experimental (HFD-EA; high-fat diet-fed rats supplemented with ellagic acid) groups are indicated as follows: * *p* < 0.05, ** *p* < 0.01.

**Figure 2 nutrients-13-00804-f002:**
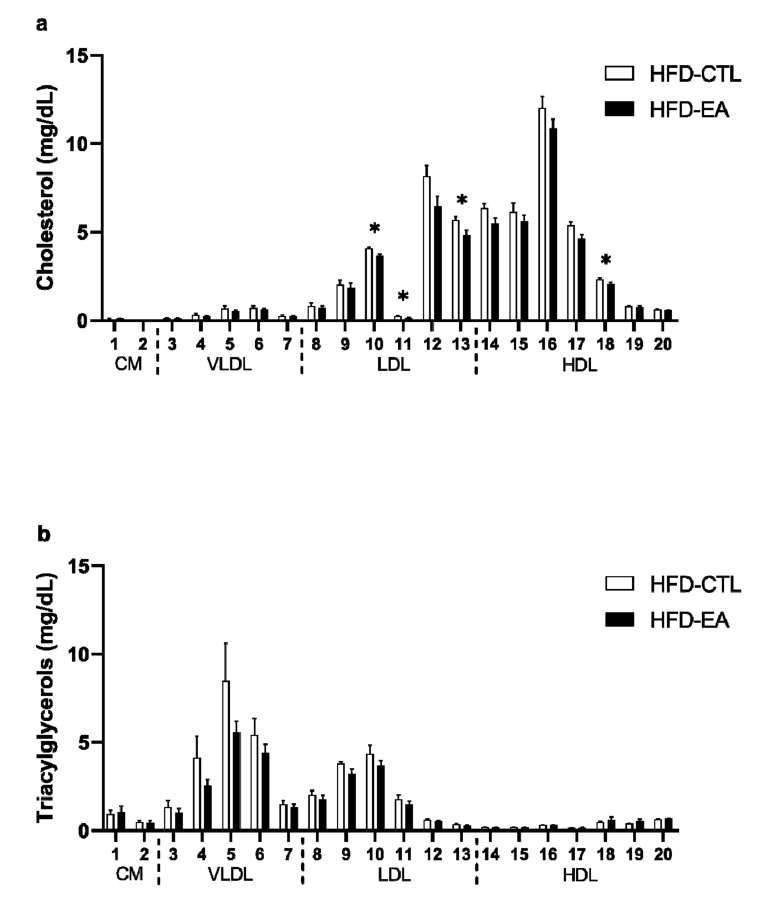
Effect of ellagic acid supplementation on cholesterol and triacylglycerol profile: (**a**) cholesterol and (**b**) triacylglycerols content in 20 lipoprotein subfractions in control (HFD-CTL; high-fat diet-fed rats) and experimental (HFD-EA; high-fat diet-fed rats supplemented with ellagic acid) groups of rats. Values are mean ± SEM, *n* = 4/group. The significance levels for comparison between HFD-CTL and HFD-EA groups is indicated as follows: * *p* < 0.05. The allocation of individual lipoprotein subfractions to major lipoprotein classes is shown in order of particle’s decreasing size from left to right. HFD-CTL, high-fat diet-fed rats; HFD-EA, high-fat diet-fed rats supplemented with ellagic acid; CM, chylomicron; VLDL, very low-density lipoprotein; LDL, low-density lipoprotein; HDL, high-density lipoprotein.

**Figure 3 nutrients-13-00804-f003:**
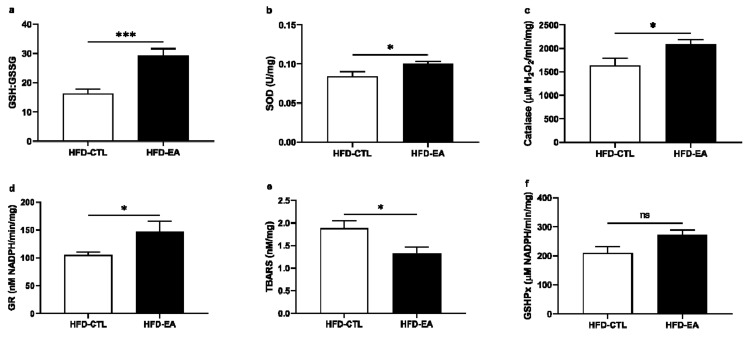
Effect of ellagic acid supplementation on oxidative stress level in the liver tissue: (**a**) reduced:oxidized glutathione ratio (GSH:GSSG); (**b**) superoxide dismutase (SOD); (**c**) catalase; (**d**) glutathione reductase (GR); (**e**) thiobarbituric acid reactive substances (TBARS); (**f**) glutathione peroxidase (GSHPx). Values are mean ± SEM, *n* = 8/group. The significance levels for comparison between control (HFD-CTL; high-fat diet-fed rats) and experimental (HFD-EA; high-fat diet-fed rats supplemented with ellagic acid) groups are indicated as follows: * *p* < 0.05, *** *p* < 0.001, ns: not significant.

**Figure 4 nutrients-13-00804-f004:**
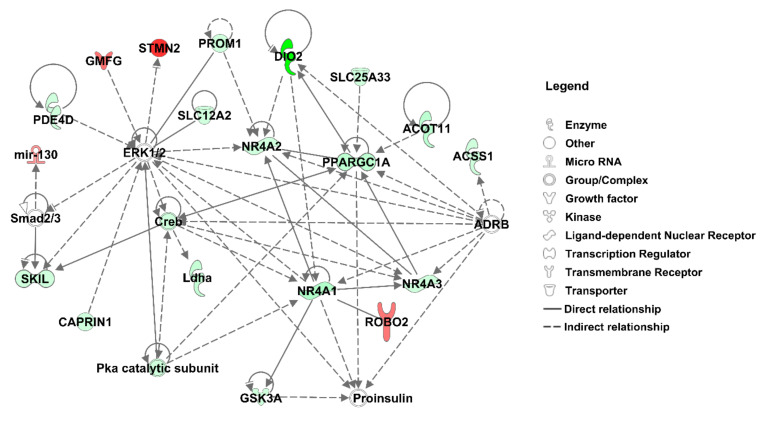
Mechanistic network reaching the highest score for the effect of ellagic acid on the expression profile of brown adipose tissue. The effect on expression of genes significantly differentially expressed between ellagic acid-fed and control rats is shown in shades of green (downregulation by EA) or red (upregulation by EA). Derivation of the network was performed using Ingenuity Pathways Analysis (Qiagen).

**Table 1 nutrients-13-00804-t001:** Effect of ellagic acid supplementation on morphometric and metabolic variables in HFD-CTL and HFD-EA rats.

Variables	HFD-CTL	HFD-EA	*p*-Value (*t*-Test)
Morphometric variables			
Initial body weight (g)	393 ± 8	380 ± 9	0.32
Final body weight (g)	412 ± 8	400 ± 9	0.35
Heart (g/100g b.wt.)	0.41 ± 0.01	0.49 ± 0.04	0.15
Liver (g/100g b.wt.)	2.91 ± 0.04	2.95 ± 0.09	0.68
Kidneys (mg/100g b.wt.)	706 ± 8	715 ± 8	0.47
Brown fat (mg/100g b.wt.)	76 ± 5	126 ± 10	0.001 *
Epididymal fat (g/100g b.wt.)	1.08 ± 0.04	0.94 ± 0.04	0.040 *
Retroperitoneal fat (g/100g b.wt.)	1.37 ± 0.07	1.11 ± 0.1	0.066
Metabolic variables			
Free glycerol (mg/dL)	1.60 ± 0.11	1.16 ± 0.10	0.033 *
Total cholesterol (mg/dL)	56.8 ± 1.5	49.5 ± 1.9	0.039 *
Total triacylglycerols (mg/dL)	37.5 ± 4.2	30.1 ± 2.0	0.21
Nonesterified fatty acids (mmol/L)	0.66 ± 0.03	0.64 ± 0.02	0.69
Alanine aminotransferase (µkat/L)	1.17 ± 0.04	1.18 ± 0.05	0.92
Aspartate aminotransferase (µkat/L)	3.57 ± 0.1	3.43 ± 0.09	0.49

Variables are mean ± SEM, *n* = 6–8 for each group. HFD-CTL, high-fat diet-fed rats; HFD-EA, high-fat diet-fed rats supplemented with ellagic acid. * indicates statistically significant *p*-values.

## Data Availability

The microarray data generated and analyzed during the current study are available in the ArrayExpress repository (https://www.ebi.ac.uk/arrayexpress/) under accession number E-MTAB-8928.

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
