# Peer review of "Ellagic Acid Affects Metabolic and Transcriptomic Profiles and Attenuates Features of Metabolic Syndrome in Adult Male Rats"

_nutrients, 2021, doi:10.3390/nu13030804_

Round 1

Reviewer 1 Report

The paper entitled "Ellagic acid affects metabolic and transcriptomic profiles and attenuates features of metabolic syndrome in adult male rats" have some major concerns that deeply affect its relevance and scientific interest:

1) Three weeks of HFD. Do the authors really think that this could be relevant to achieve a metabolic syndrome state?

2) Absence of control (no HFD) group. Which is the real impact of these 3 weeks (HFD)??? Unknown.

3)Which is the rationalle of using the SHR-Zbtb16Lx/k.o. strain?? Which is its real relevance in the discusion and conclusion? Unknown.

4) The effect of ellagic acid in brown adipose tissue is not new.

Due to these important flaws I think that the paper is not suitable for its publication in its present form.

Author Response

RESPONSE TO COMMENTS OF REVIEWER 1

The paper entitled "Ellagic acid affects metabolic and transcriptomic profiles and attenuates features of metabolic syndrome in adult male rats" have some major concerns that deeply affect its relevance and scientific interest:

Thank you for thoroughly reviewing our manuscript and pointing out relevant issues. In the revised version of the manuscript, we have addressed all of them as shown in detail below.

1) Three weeks of HFD. Do the authors really think that this could be relevant to achieve a metabolic syndrome state?

The protocol of the study indeed involved short-term HFD feeding. We concur that in the general diet-induced models of metabolic syndrome, this timeframe would often be insufficient for inducing the full metabolic syndrome. However, as we specifically selected an inbred model carrying alleles repeatedly shown to greatly sensitize their carrier to dietary and pharmacologic interventions, becoming apparent even after two weeks (e.g. Physiol Genomics. 2005 Apr 14;21(2):243-52., Folia Biol. 2005;51(3):53-61), we believe that the selected duration of HFD is sufficient to elicit shifts towards the metabolic syndrome-related features. Similar studies also employed short-term HFD exposure - Ann Nutr Metab 2011;58:220–223, Appl Physiol Nutr Metab . 2017 Feb;42(2):181-192., Food Chem Toxicol. 2009 Jan;47(1):50-4., Phytomedicine . 2017 Dec 15;37:4-9). However, the aim of our study was not to assess the HFD effect as this factor did not differ between control and experimental groups (please see answer below regarding the study design).

2) Absence of control (no HFD) group. Which is the real impact of these 3 weeks (HFD)??? Unknown.

Thank you for the comment. In our protocol, we were comparing two groups of genetically identical animals kept in the same environment, fed the same diet (HFD) with the only distinct component of the protocol being the administration of the ellagic acid. We concur that there are several other possible valid experimental designs addressing different questions, i.e. standard chow group, standard chow group + ellagic acid, HFD, HFD + EA, high-carbohydrate diets, different dosages of ellagic acid, different timelines, different ages of intervention etc. and we acknowledged that in the list of limitations of our study. In our study, we focused solely on the EA effect in the genetically susceptible model fed identical diet as the control group. This approach is relatively common in literature, including studies on EA e.g. Ding Y, Zhang B, Zhou K, Chen M, Wang M, Jia Y, Song Y, Li Y, Wen A. Dietary ellagic acid improves oxidant-induced endothelial dysfunction and atherosclerosis: role of Nrf2 activation. Int J Cardiol. 2014 Aug 20;175(3):508-14.; or other nutraceuticals -Jang, M.H., Mukherjee, S., Choi, M.J. et al. Theobromine alleviates diet-induced obesity in mice via phosphodiesterase-4 inhibition. Eur J Nutr 59, 3503–3516 (2020); Hoek-van den Hil, E.F., van Schothorst, E.M., van der Stelt, I. et al. Quercetin decreases high-fat diet induced body weight gain and accumulation of hepatic and circulating lipids in mice. Genes Nutr 9, 418 (2014). On the other hand, the effects of administration of short- and long-term diets rich in carbohydrates or fats in genetically designed models (including those with specific Zbtb16 alleles) is certainly of major interest to us and we are pursuing the relevant comparisons in other experiments currently underway.

3) Which is the rationalle of using the SHR-Zbtb16Lx/k.o. strain?? Which is its real relevance in the discusion and conclusion? Unknown.

As pointed out above, we based our selection of model strain on previous results of our group and others showing the involvement of distinct Zbtb16 alleles in the metabolic syndrome pathogenesis both in experimental models and humans, as well as in specific nutrigenetic and pharmacogenetic interactions affecting the distinct aspects of the syndrome. By utilizing a genetically unique strain, we add to the variety of other models subjected to EA administration, modelling, in part, the genetic heterogeneity of the human population, where carriers of distinct sensitizing allele combinations might react in an idiosyncratic fashion to the administered nutraceutical. In the revised manuscript, we clarified the rationale for the selection of our model system.

4) The effect of ellagic acid in brown adipose tissue is not new.

We concur that the effects of EA on the brown adipose tissue were described before and so we did not claim novelty in this regard, as we provided the references to prior papers. That said, we are not aware of works addressing the EA effects on the brown adipose tissue at transcriptome level.

Reviewer 2 Report

Dear authors,

the article entitled "Ellagic Acid Affects Metabolic and Transcriptomic Profiles and Attenuates Features of Metabolic Syndrome in Adult Male Rats" presents results of a chronic study carried out with animals supplemented with Elagic acid. The manuscript is well written and clear and the results are pretty interesting. My only concern is about the duration of the experiment. Is 3 weeks of HFD enough to develop metabolic syndrome in the animals? This should be considered  and comment in the discussion section. 

Other minor comments are the following:

Line 13: Rather than call it "solvent" I'd say "vehicle"

Lines 20-22: Which is the metabolic relevance of these altered genes in relation with metabolic syndrome? These two sentences should be more linked in order to make clear the message.

Line 109: 12 h is in red.

Lines 124-125. Procedure followed to assess lipid profile is not clear enough. Authors should clarify and rewrite it.

In Table 1 the name of the last column is not clear enough. Authors may call it "p-value (t-test)", for example.  In additon, significant values could be highlighted with a symbol (*) so readers would rapidly see the differences. Decimals should be marked with dots. (2.95 + 0.09) 

Figure 1: "ns" is not used so it should be removed from figure caption.

Line 219: Authors should cite Table 1 when refeering to these data.

Line 224: same as in the other comment regarding ALT and AST

Figure 2 caption: neither **, *** nor ns are used so they should be deleted from the caption

Line 234: "online resource 1" I guess that authors refeer to the Supplementary Table S1. If so, it should be cited appropriately. The same applies for the citation of the other online resources.

Line 236: Figure 3 only shows the results of oxidative stress markers in liver. Kidney data is not presented here. Therefore, authors should delete this from the text.

Lines 303-306. Authors claim that a decrease of serum resistin could be mediating the results observed in insulin. However the did not directly analyse this and are basing their theory in others results. Could they support this with their own data? For instance the transcriptome of PCR results? If it's not possible to do this or complement them with new data they should be more clear and discuss it the text.

Line 357: It should be corected by UCP1

Reviewer 3 Report

They aim to analyze metabolic and transcriptomic responses to ellagic acid treatment in a genetic rat model of metabolic syndrome. They assess a transcriptomic profile of liver, brown, and epididymal adipose tissues and also several morphometric and metabolic parameters in adult male rats, carrying a variant of one of the metabolic syndrome-related genes, fed a high-fat diet accompanied by daily intragastric gavage of ellagic acid.

It is very complex study with an experimental design that wants to attain very diverse objectives driven to obtain too ambiguous conclusion.

Minor:

Materials & Methods

Lines 112-113: (“Blood glucose concentrations over the period of 180 minutes were used to calculate the area under the curve. All rats were then sacrificed… “) Are the rats sacrificed after ending the glucose tolerance test?

Lines 129-132: No precaution was taken to avoid the oxidation of glutathione during sampling and processing the tissue.

Lines 178-182: Why not one-way ANOVA statistic analysis in place of unpaired t-student test?

Results

Table 1: What HFD-CTL x HFD-EA means? The metabolic results shown in table1 do not comment in the text.

Figure 1: The legend does not indicate between which groups are the symbols *, **, *** significant differences. Example: The figure a indicate the symbol ** on the values of blood glucose at 0 minutes. What means it?

Lines 219-224: It comments the results of table 1 as it was presented in Figure 2. However, no comments about the results shown in the Figures 2a and 2b has been exposed.

Figure 3: Figure 3 shows oxidative stress markers in the liver tissue, however the text (235-237) tell about the liver, kidneys and heart tissues. Can you clarify it?

Discussion

Lines 287-289: A decrease in the relative weight of epididymal fat was not determined. The experimental design allows observing a lower increase in epididymal fat of rats fed with high fat diet plus Elagic Acid than the rats fed with high fat diet plus solvent. The decrease in the relative weight point to a slimming effect, what is not the case.

Lines 296-321: These results obtained in this study have not novelty and it does not provide   new evidences about the possible mechanisms implied.

Lines 336-338: These sentences are speculative.

Lines 363-364: Brown fat did not increase relative weight; in fact, the brown fat relative weight at the begin of high fat diet was unknown both in the HFD-CTL and HFD-EA groups.

Lines 375-376: The mechanistic network for the effect of ellagic acid on the expression profile of brown adipose tissue seems to show a different profile as indicated in the lines 375-376 ‘the brown adipose tissue does not seem to be metabolically activated by ellagic acid in the SHR-Zbtb16Lx/k.o. strain but rather used as a preferential excess fat depot’.

Conclusion: The conclusions are too general, probably the highest number of variables determined does not allow them to reach a valid conclusion in this sense.

Author Response

RESPONSE TO COMMENTS OF REVIEWER 3

 They aim to analyze metabolic and transcriptomic responses to ellagic acid treatment in a genetic rat model of metabolic syndrome. They assess a transcriptomic profile of liver, brown, and epididymal adipose tissues and also several morphometric and metabolic parameters in adult male rats, carrying a variant of one of the metabolic syndrome-related genes, fed a high-fat diet accompanied by daily intragastric gavage of ellagic acid.

It is very complex study with an experimental design that wants to attain very diverse objectives driven to obtain too ambiguous conclusion.

Thank you for thoroughly reviewing our manuscript and pointing out relevant issues. In the revised version of the manuscript, we have addressed all of them as shown in detail below.

Minor:

Materials & Methods

Lines 112-113: (“Blood glucose concentrations over the period of 180 minutes were used to calculate the area under the curve. All rats were then sacrificed... “) Are the rats sacrificed after ending the glucose tolerance test?

We have clarified the description of the protocol steps in the revised version of the manuscript.

Lines 129-132: No precaution was taken to avoid the oxidation of glutathione during sampling and processing the tissue.

Thank you for the comment. For all analysis of oxidative stress markers, after removal tissue samples were frozen immediately in liquid nitrogen and stored at -80°C. During analysis tissue homogenate were processed on ice. For analytical reasons, no chemical additive such as BHT was added.

Lines 178-182: Why not one-way ANOVA statistic analysis in place of unpaired t-student test?

We believe that in case of comparing only two groups, the two statistical approaches are mathematically equivalent (unless the t-test would use a correction for unequal variances, which was not the case of our study) – F-test is the generalization of the t-test. We have re-analyzed all the comparisons using one-way ANOVA and retrieved identical sets of p-values.

Results

Table 1: What HFD-CTL x HFD-EA means? The metabolic results shown in table1 do not comment in the text.

We have modified the heading of the last column of Table 1 to “p-value (t-test)” to clarify the meaning. In the revised version of the manuscript, the Table 1 is referred to in the relevant parts of the text.

Figure 1: The legend does not indicate between which groups are the symbols *, **, *** significant differences. Example: The figure a indicate the symbol ** on the values of blood glucose at 0 minutes. What means it?

Thank you for noting this, we have provided a more detailed description of the significance levels and the compared groups in the legends of all relevant Figures. In general, the symbols correspond to the significance level of comparison between control (HFD-CTL; high-fat diet-fed rats) and experimental (HFD-EA; high-fat diet-fed rats supplemented with ellagic acid) - * p < 0.05, ** p < 0.01, *** p < 0.001 etc. Therefore, the ** symbol over the glucose at 0 minutes indicates the statistically significant difference (p < 0.01) between fasting glucose of HFD-CTL and HFD-EA.

Lines 219-224: It comments the results of table 1 as it was presented in Figure 2. However, no comments about the results shown in the Figures 2a and 2b has been exposed.

We have modified the text describing the mentioned results so all Tables and Figures are adequately quoted and the related results described.

Figure 3: Figure 3 shows oxidative stress markers in the liver tissue, however the text (235-237) tell about the liver, kidneys and heart tissues. Can you clarify it? Thank you for noting this omission, we have rephrased the relevant portion of the text to reflect only the results of oxidative stress assessment in the liver.

Discussion

Lines 287-289: A decrease in the relative weight of epididymal fat was not determined. The experimental design allows observing a lower increase in epididymal fat of rats fed with high fat diet plus Elagic Acid than the rats fed with high fat diet plus solvent. The decrease in the relative weight point to a slimming effect, what is not the case.

We have rephrased the Discussion to correspond with the suggestion.

Lines 296-321: These results obtained in this study have not novelty and it does not provide new evidences about the possible mechanisms implied.

We agree that several effects of ellagic acid administration in our model are in agreement with previous results, however, we believe that even these should be discussed in the light of strain-specific combination of EA effects.

Lines 336-338: These sentences are speculative.

We have removed the sentence in the revised version of the manuscript.

Lines 363-364: Brown fat did not increase relative weight; in fact, the brown fat relative weight at the begin of high fat diet was unknown both in the HFD-CTL and HFD-EA groups.

We have rephrased the Discussion to correspond with the suggested notion.

Lines 375-376: The mechanistic network for the effect of ellagic acid on the expression profile of brown adipose tissue seems to show a different profile as indicated in the lines 375-376 ‘the brown adipose tissue does not seem to be metabolically activated by ellagic acid in the SHR-Zbtb16Lx/k.o. strain but rather used as a preferential excess fat depot’.

We believe that based on the substantial decrease in brown adipose tissue-enriched gene Ppargc1a as well as of brown adipose tissue activation markers Dio2 and Nr4a1 as indicated in the mechanistic network together with other differentially expressed genes (Supplementary Table 2) support the notion expressed in the manuscript.

Conclusion: The conclusions are too general, probably the highest number of variables determined does not allow them to reach a valid conclusion in this sense.

We have reformulated the conclusions to reflect closer the presented data.

Round 2

Reviewer 1 Report

The authors have fully addressed the suggested coments 

Author Response

Thank you again for your insight and comments that helped to improve the manuscript, we appreciate that

Reviewer 3 Report

All minor issues have been corrected and clarified and the wording of the conclusion has been improved. The revised version of the manuscript is acceptable for publication. However, the lack of addition of glutathione oxidation protectors in the samples alters the possible value of the GSH: GSSG ratio as an indicator of oxidative stress. This aspect would have to be indicated in the text of the published version.

Author Response

Thank you for your comment. We have included the statement in the limitations section of the Discussion in the revised manuscript.